# Dynamics and diversity in adolescents' experienced barriers and facilitators for physical activity maintenance

Timothy A. Houtman[1]*, Froukje Sleeswijk Visser[1], Amy van Grieken[2], Valentijn Visch[1]

1 The Human-Centered Design Department, Industrial Design Engineering, Delft University of Technology, Delft, the Netherlands, 2 The Department of Public Health, Erasmus Medical Center, Rotterdam, the Netherlands

* T.A.Houtman@TUDelft.nl

## Abstract

Despite the well-documented health benefits of physical activity (PA), over 80% of adolescents worldwide fail to meet recommended daily levels. This study identified experienced barriers and facilitators for PA maintenance among Dutch adolescents, examine how they form barrier and facilitator profiles, and explore how barriers and facilitators evolve over time. First, we conducted 21 interviews with adolescents (13–16 years) to uncover relevant barriers and facilitators. Then, we developed and applied a card sorting task based on Q methodology, and examined barrier and facilitator configurations with 30 adolescents (aged 13–18 years) who had maintained a physical activity for ≥2 years, followed by interviews. Factor analysis revealed five facilitator profiles: *Mental and physical health benefits*, *A way to be myself around others*, *Pursuing health goals*, *Developing confidence and strength*, and *Developing along my own path*. Four barrier profiles emerged: *Low motivation and energy*, *Not a good fit for me*, *Balancing act*, and *Proximity, possibility and perception*. Facilitator profiles ranged from immediate characteristics of the activity, such as enjoyment, social connection, and mental well being, to more future-oriented drivers such as autonomy and self-development. Barriers profiles varied from predominantly internal (e.g., low motivation) to external (e.g., distance, weather) or mixed influences linking life demands to reduced personal resources. Across participants, enjoyment was the most consistent facilitator, but perceived influences often shifted with time, from immediate, activity-based facilitators related to competence and relatedness, toward motivations tied to autonomy, identity and coping with growing responsibilities characteristic of this life stage. These results highlight the diversity and dynamic nature of barriers and facilitators in adolescent PA maintenance. Tailoring PA promotion programs to adolescents' evolving motivations and constraints can increase their effectiveness, supporting sustained active lifestyles into adulthood.

**Data availability statement:** Data cannot be shared publicly because no informed consent was obtained from participants and their guardians for sharing their data publicly, even in anonymized form. Data are available from 4TU.ResearchData (contact via https://data.4tu.nl/) for researchers who meet the criteria for access to confidential data.

**Funding:** This research was funded by a grant awarded to dr. V.T. Visch from the Dutch Research Council (NWO). File Number: 403.19.246.

**Competing interests:** The authors have declared that no competing interests exist.

Despite well-documented physical and psychological benefits, such as cardiovascular health and stress regulation [1,2], over 80% of adolescents worldwide fail to meet daily PA recommendations [3]. Specifically, 77.6% of boys and 84.7% of girls aged 11–17 are insufficiently active, and activity levels often drop further as they transition into adulthood [3–6]. Although many interventions can increase PA levels during the intervention period these short-term gains typically erode within six to twelve months [7–10], highlighting the need to understand what enables and hinders long-term maintenance of healthy behaviors in adolescence.

Current PA promotion programs tend to emphasize intervention outcomes, often focusing on risk-reduction metrics such as BMI improvement and minutes of moderate-to-vigorous activity [11]. Although these evidence- and theory-based interventions (e.g., using behavior-change techniques) can produce short-term PA increases, a key limitation is that their design tends to be static and is not adapted to the rapid physical, psychological and social changes that characterize this life stage.

A person-based, developmental approach offers an alternative by treating adolescents as active agents whose goals and constraints can change over time. Drawing from self-determination theory [12] and developmental science, the person-based, developmental approach considers adolescents' subjective experiences and values as well as how their motives and barriers shift throughout their development [11]. As adolescents explore new social roles and identities, their motivations and social surroundings change and become increasingly diverse. Given the strong links between intrinsic motivation, identity congruence, and long-term PA maintenance [13,14], interventions that incorporate adolescents' own experiences are more likely to foster sustainable health behavior change. Accordingly, this paper has two aims: (1) to identify profiles of experienced barriers and facilitators for PA maintenance among adolescents and (2) to examine how experienced barriers and facilitators are perceived to change over time.

Facilitators are factors that enable or motivate a behavior, whereas barriers are factors that obstruct or demotivate it. Prior research has identified a wide array of PA facilitators and barriers including individual, social, and environmental factors [15]. For example, adolescents often cite enjoyment and social support as primary reasons for participation, while lack of time, low motivation, or access often prevent PA participation [15,16].

However, barriers and facilitators are not isolated but interact within each adolescent's life. Yet most research treats barriers and facilitators as general characteristics of adolescents, aggregated across large groups (e.g., all adolescents) [15,16] or simple subgroups (e.g., by gender) [17,18] thereby obscuring the unique configurations that drive individual PA. Because motivation and experience around PA are made up from multiple connected personal and contextual influences [19], PA behavior may be better understood as profiles of barriers and facilitators. These combinations create a more holistic picture than single indicators alone. Moreover, facilitators can reinforce one another but also compete with other life motivations. For example, Strömmer and colleagues [20] show that while adolescents tend to value their health, their motivations for engaging in health behavior, like PA, can be at odds with their

growing desires for autonomy or social belonging. This finding highlights the importance of recognizing how barriers and facilitators are interconnected and how they sometimes compete with or enhance each other in shaping adolescents' PA experiences.

Barriers and facilitators for PA not only interact within an individual's life but also evolve over time. Especially during adolescence, a developmental period marked with major life changes in which their goals and priorities can shift [21]. For example, adolescents often experience shifts in motivation and attitudes towards PA during key transitions such moving from primary to secondary school [17,22]. As they gain new experiences, they explore different interests, form new peer groups, and exercise increasing autonomy in their decision making, all of which shapes their active lifestyle trajectories [19].

Theoretical perspectives also suggest that the motives and capacities driving early PA engagement differ from those needed to maintain PA [14]. These initial facilitators must change and adapt over time to support long-term PA maintenance. Research on adolescent PA adherence shows that a passion for the activity, perceived positive outcomes, identity alignment with the activity, and continued support from friends and parents are key determinants for sustained PA engagement [17,19,22,23]. These findings aligns with behavior maintenance theories, which emphasize intrinsic motivation, identity, and habit formation (built through repeated practice in stable contexts), as essential for long-term health behavior maintenance [13]. However, these facilitators of PA maintenance may not be present in the early phases of PA engagement and need to be cultivated over time. Even among adolescent PA "maintainers", the barriers and facilitators are expected to shift over time both because of the changes associated with this developmental period and also the facilitators of PA maintenance differ from those of earlier PA engagement. Understanding these dynamics can inform how health promotion programs can support adolescents' changing barriers and facilitators.

This paper examines adolescents' experienced barriers and facilitators for PA maintenance with two aims. The first aim is to examine profiles of experienced barriers and facilitators associated with PA maintenance and how these vary across adolescents. The second aim is to investigate how PA barriers and facilitators are perceived to change during adolescence. By examining these profiles and perceived dynamics, we will gain richer insights into adolescents' experiences with PA maintenance. These findings can support the design of interventions that resonate with young people's own perspectives and aim to promote sustained PA engagement.

## Methods

We employed a mixed methods approach over two main phases to gain a rich understanding of adolescents' experienced barriers and facilitators of PA maintenance. In phase 1, we conducted semi-structured interviews to explore adolescents' perceptions and views on barriers and facilitators of PA. In phase 2, we translated these insights into a card sorting activity based on Q methodology to gain insights into personal profiles of barriers and facilitators and how they can change over time.

Q methodology combines quantitative and qualitative techniques to investigate the range of possible perspectives on a topic [24,25]. Its purpose is to uncover subjectivity by identifying the extent to which perspectives are shared or differ across people. In a Q-sort, participants rank a structured set of items, which forces prioritization and trade-offs rather than evaluating each item in isolation. These ranked configurations are then analyzed using by-person factor analysis to identify shared profiles of participants, which are then interpreted qualitatively using participants' explanations from interviews.

This approach is well suited to our aim of understanding adolescent experience as configurations of interrelated influences rather than isolated barriers or facilitators. Because participants rank the items themselves and confirm they reflect their experiences, the resulting profiles offer a holistic view of how multiple influences shape PA maintenance experiences according to adolescents. The combination of the quantitative structure with qualitative insights enables the systematic identification and interpretation of shared patterns in our sample.

## Ethics

Ethical approval for the study was obtained from the Human Research Ethics Committee of Delft University of Technology (submission number 1790). Permission was granted by four secondary schools in the Rotterdam area, the Netherlands to approach their student on school premises. After approval from a designated school staff member, a digital information letter was sent to parents and guardians at least one week before approaching participants. The letter outlined the study and explained that their child might be approached on school premises. On the day of data collection, researchers approached students during free periods (breaks, mentor classes, or after school), explained the study verbally, and obtained written informed consent. Each participant received a physical follow-up letter to their parents confirming their participation and reminding the parents about the study and the right to withdraw their child. Participants were also given a €5 cinema voucher for their participation. Only non-identifying demographic data (age and gender) was recorded. The consent procedure was developed in collaboration with the legal department of Delft University of Technology.

## Recruitment

The data collection for phase 1 took place in October 2021 the data collection on phase 2 took place the second half of 2022. With approval from the school, the first author and a research assistant (acknowledgements) visited each school. In phase 1, we invited any student interested in sharing their views on health and PA, in phase 2, we approached students who had engaged in a specific physical activity for at least two years. Students were approached during gap hours, breaks, after school, or, when arranged, during mentor class sessions with permission of the teacher.

## Phase 1: Exploratory interviews

To identify barriers and facilitators relevant to our local target group, we conducted interviews with 21 adolescents aged 13–16 ($M_{age}$ = 14.2). During the interviews, we used generative methods to elicit personal experiences around health behavior [26]. We used objects and visuals (S1 Fig) were used to engage participants and prompt discussion about what an active lifestyle means to them. For example, we presented illustrated characters and asked participants to indicate which they perceived as the most and least fit. We also asked them to rank various objects (e.g., a ball for exercise, a toy bed for sleep, fruit for food) according to their importance for the participants' fitness. These objects and images, selected by the researchers, were intended to evoke different dimensions of health (exercise, diet, sleep, etc.). Furthermore, we asked participants the barriers and facilitators they have faced when trying to change or maintain their health behavior. Their choices and statements were written down and the interviews lasted approximately 30 minutes.

## Phase 2: Card sorting

Based on the insights of the interviews we developed a card sorting activity based on Q methodology. Following this method, we asked participants to rank a series of statement cards, called a Q-set, about the topic according to their personal views. This ranking process, called Q-sorting, allowed us to understand the subjective viewpoints of the participants. Consequently, participants were interviewed about their Q-sorts. The Q-sorts were then used to perform a by-person factor analysis revealing shared perspectives on the topic [27].

**Step 1. Q-set development.** Based on the exploratory interviews, two Q-sets were developed: one for barriers and one for facilitators. The interview results were analyzed using the barrier and facilitator themes identified by Martins and colleagues as a guide [16], which resulted in 41 barriers and 49 facilitators. The two lists were reviewed within the research team and the redundancy between statements was reduced by removing ones with overlapping themes, resulting in two balanced Q-sets. The two sets consisted of 31 barriers and 36 facilitators (S2 Table).

**Step 2. Q-sorting.** Q-sorting was conducted with 30 adolescents aged 13–18 ($M_{age}$ = 15.1, 16 male). Each Q-sorting lasted approximately 30 minutes and consisted of asking each participant to:

1. select a physical activity that they had maintained for at least two years and are still doing currently.

2. respond to the facilitator Q-set statements, which were shown one by one, and to categorize the statements as 'not important', 'a little important' or 'very important' for maintaining their physical activity.

3. select the top 10 from the 'very important' stack and sort them in the ranking structure based on personal importance (Fig 1).

4. confirm that they were satisfied with the configuration and that it reflected their experience, before explaining each sorting choice they had made and its role in their PA maintenance.

5. reflect on any changes in their facilitators over time, and if applicable, re-sort them in line with their experiences from two years ago

6. explain how and why their facilitators had changed over time.

7. repeat steps 2–6 with the Q-set about PA barriers.

**Step 3. Analyses.** Using the data obtained in step 2, we investigated the descriptive statistics to identify the most important barriers and facilitators based on cumulative ranked scores for all adolescents in our sample. The types of PA that adolescents had maintained can be seen in S3 Table.

We then conducted a by-person factor analysis (PCA involving varimax rotation) of the facilitator and barrier Q-sorts using the *qmethod* package in R (version: 1.3.1093) [28,29]. The number of factors was determined using visual inspection of the screeplot, the Eigen-value criteria > 1, and at least 3 participants per factor which are commonly used criteria for this method [27]. Next, we developed weighted average z-score tables for displaying the ideal Q-sort and focused our interpretation of the factors, which we will also call profiles, on the statements ranked +2 or above, as determined by these tables (S4 Table).

Perceived changes over time were examined by calculating the change in rank order score of each barrier and facilitator for participants who indicated a change. These rank order changes were summed for each barrier and facilitator to identify the influences that shifted most between participants' past and current PA experiences.

The profiles and largest retrospective shifts were analyzed and interpreted qualitatively, utilizing participants' explanations and anecdotes about their choices, and guided by the themes described by Martins. Participant quotes were visually

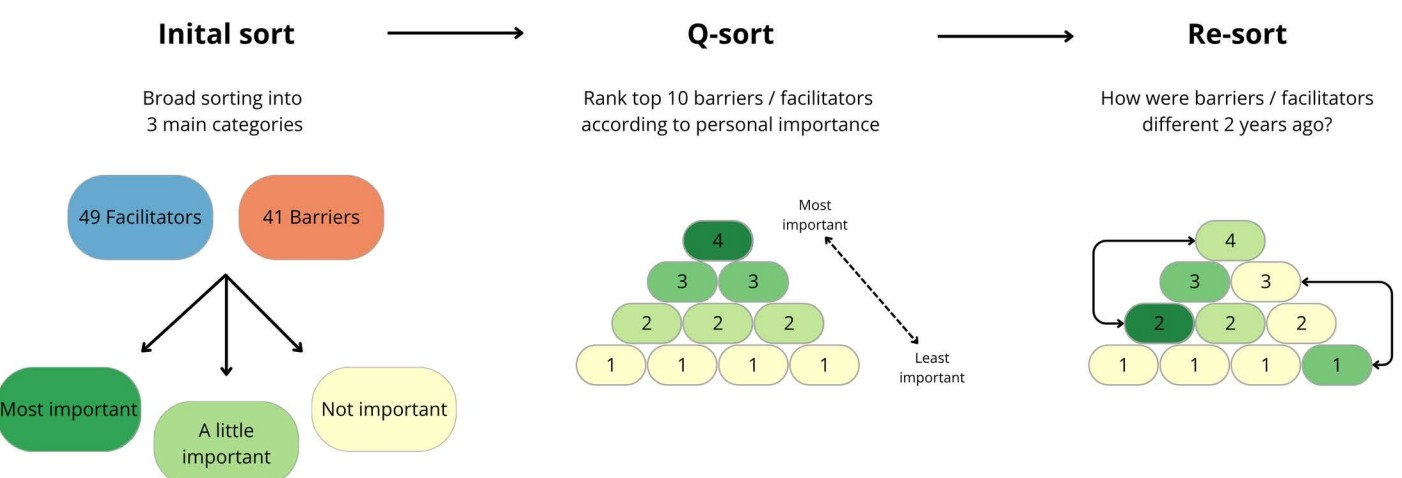

**Fig 1. Research protocol during Q-sorting.**

organized on statement cards within the relevant profiles and for the largest barriers and facilitators changes. Throughout the analysis, researchers stayed closely aligned with participants' own words to ensure a faithful representation of their experiences.

## Results

### Descriptive statistics

Table 1 presents the overall highest scoring facilitators and barriers of the group based on rank scores. Results suggest that having fun was the most important facilitator. The most important barriers were related to having other responsibilities or too little time or more related to personal factors such as having too much on one's mind and too little energy.

### Factor analysis

**Facilitator profiles.** Two to seven factors were considered, and a five-factor solution was selected based on our criteria as described in the methods section. The final five factors included 20 (out of 30) participants and explained 53.7% of the total variance each representing a distinct profiles (see Fig 2).

### Profile A – "Mental and physical health benefits"

These adolescents placed significant value on the mental and physical advantages of engaging in physical activity, while emphasizing the importance of enjoying these activities with others. For these individuals, participation in physical activity served as a coping mechanism for managing stress and energy, enabling them to clear their mind.

### Profile B – "A way to be myself around others"

This factor of adolescent girls placed great value on authenticity, fun, and self-expression in their engagement with PA. Participants emphasized the significance of being able to be their authentic self when doing physical activities in social groups, which significantly contributed to their enjoyment of the activity.

### Profile C – "Pursuing health goals"

These adolescents were driven by goal achievement, such as earning diplomas and acquiring specific skills, and maintaining their physical fitness for their future wellbeing. They seek to grow personally through facing challenges while also working on their overall health and wellbeing. Overall, they regarded their PA as a fundamental component of a well balanced lifestyle.

**Table 1. The total scores of most important barriers and facilitators in our sample, n = 30.**

| Facilitators | Type | Total ranked score (out of possible 120) |
|---|---|---|
| Having fun | Activity | 60 |
| Clearing the mind | Individual | 36 |
| Being able to be yourself | Individual | 33 |
| Wanting to maintain something | Life | 29 |
| **Barriers** | | |
| Other responsibilities | Life | 45 |
| Too little energy | Individual | 44 |
| Too little time | Life | 41 |
| Having too much on one's mind | Individual | 38 |

 

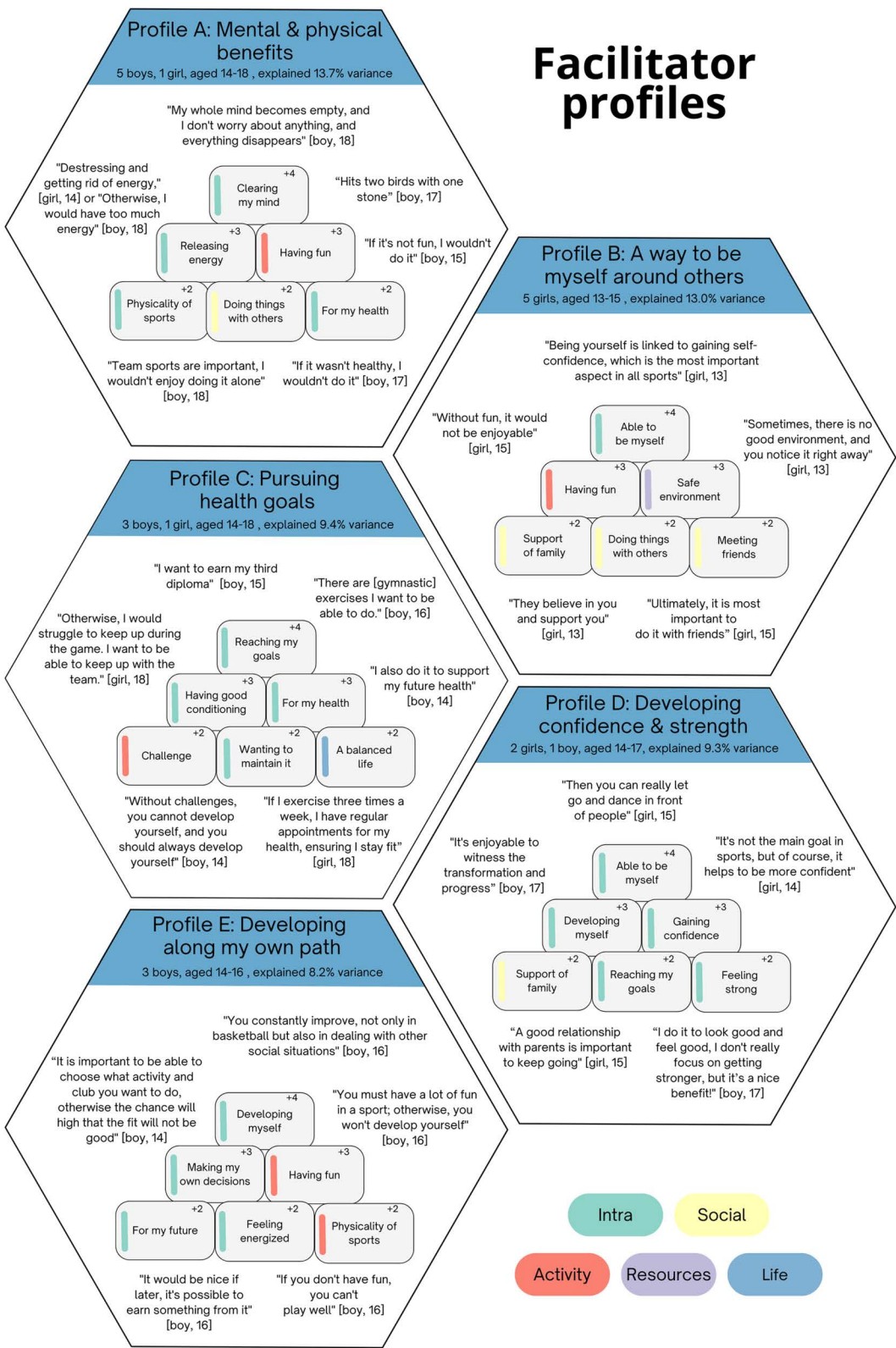

**Fig 2. Identified facilitator profiles for physical activity maintenance.**

### Profile D – "Developing confidence and strength"

These adolescents articulated the significance of personal growth and self-identity, while also recognizing the positive impact of physical activity on their overall wellbeing. A central aspect was the value placed on the importance of expressing their true selves openly, even in public settings. Personal development held considerable importance, with individuals finding joy in their own transformative journeys and progress.

### Profile E – "Developing along my own path"

The adolescent boys in this profile emphasized personal development, not only through progress in their chosen activity but also by enhancing their ability to handle other life situations. They place great importance on autonomy, valuing the freedom to make their own choices, for example, in selecting of their preferred activity or club. Enjoyment was also a significant influence, with participants recognizing fun as essential to both performance and personal growth.

**Barrier profiles.** Three to five factors were considered, and the four-factor solution was selected based on our criteria. The four factors included 27 participants and explained 53% of the total variance each representing a distinct profile (see Fig 3).

### Profile A – "Low motivation and energy"

Adolescents in this profile expressed difficulties such as feeling physically or mentally unwell, which hindered their motivation for PA. Additionally, lack of time and the burden of other obligations like school and work occupied their minds, further impeding their enthusiasm to participate in physical activities.

### Profile B – "Not a good fit for me"

These adolescents struggled most with the at times physically and emotionally unsafe environments in which their PA took place. Time constraints posed a significant challenge for this group, as they strived to balance their school commitments with engaging in physical activities. The compatibility of the activity with their preferences was also crucial, as it boosted their enjoyment and motivation to participate regularly. Financial burdens of the activity also played a role, arising from, for example, regular contributions and the cost of necessary equipment.

### Profile C – "Balancing act"

These adolescents attributed their challenges to multiple competing responsibilities particularly involving schoolwork and part-time jobs, which often take sometimes take priority over engaging in physical activities. Falling out of their exercise routine was a recurring occurrence among this group, and they sometimes struggle to get back on track after moments of laziness. Distractions, such as mobile phones or other leisure activities, also interfered with their physical activity plans.

### Profile D – "Proximity, possibility and perception"

These adolescents faced challenges in overcoming environmental barriers, particularly distance, adverse weather conditions, and competing commitments such as school obligations, which hindered their engagement in physical activities. Feelings of discomfort exacerbate their reluctance to participate, as they were concerned about their performance level and the fear of judgement from others.

### Perceived changes over time

After the initial sorting of the cards, participants were asked whether and how their barriers and facilitators for the same activity had changed compared to two years earlier. 21 out of 30 (70%) adolescents indicated that their facilitators had changed over time (13 male, 8 female, ages 13–18). 17 out of 30 (56.7%) indicated changes in their barriers (10 male, 7 female, ages 13–18). The most notable shifts, which were shared by at least four participants, are shown in Fig 4.

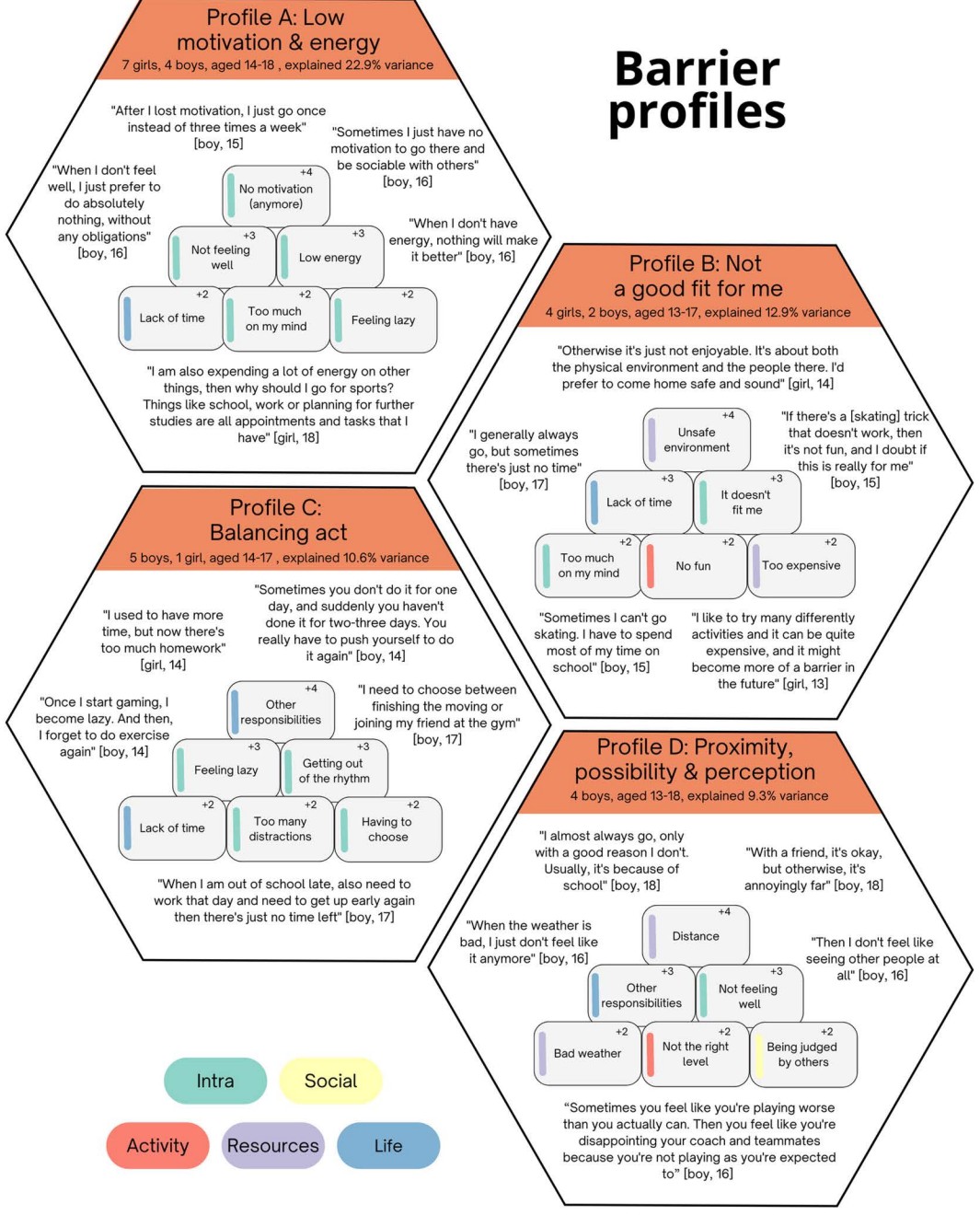

**Fig 3. Identified barrier profiles for physical activity maintenance.**

Participants who reported changes in either barriers or facilitators were slightly older on average (15.3 years) than to non-changers (14.3 years). The age range for changers was also slightly wider (ages 13–18) compared to non-changers (ages 13–16). Those who reported changes in both barriers and facilitators were the oldest on average (15.7, n = 14), compared to those reporting changes in facilitators only (15.1 years, n = 7) or in barriers only (14.0 years, n = 3). These patterns suggest that older participants were more likely to report changes in their perceived barriers and facilitators.

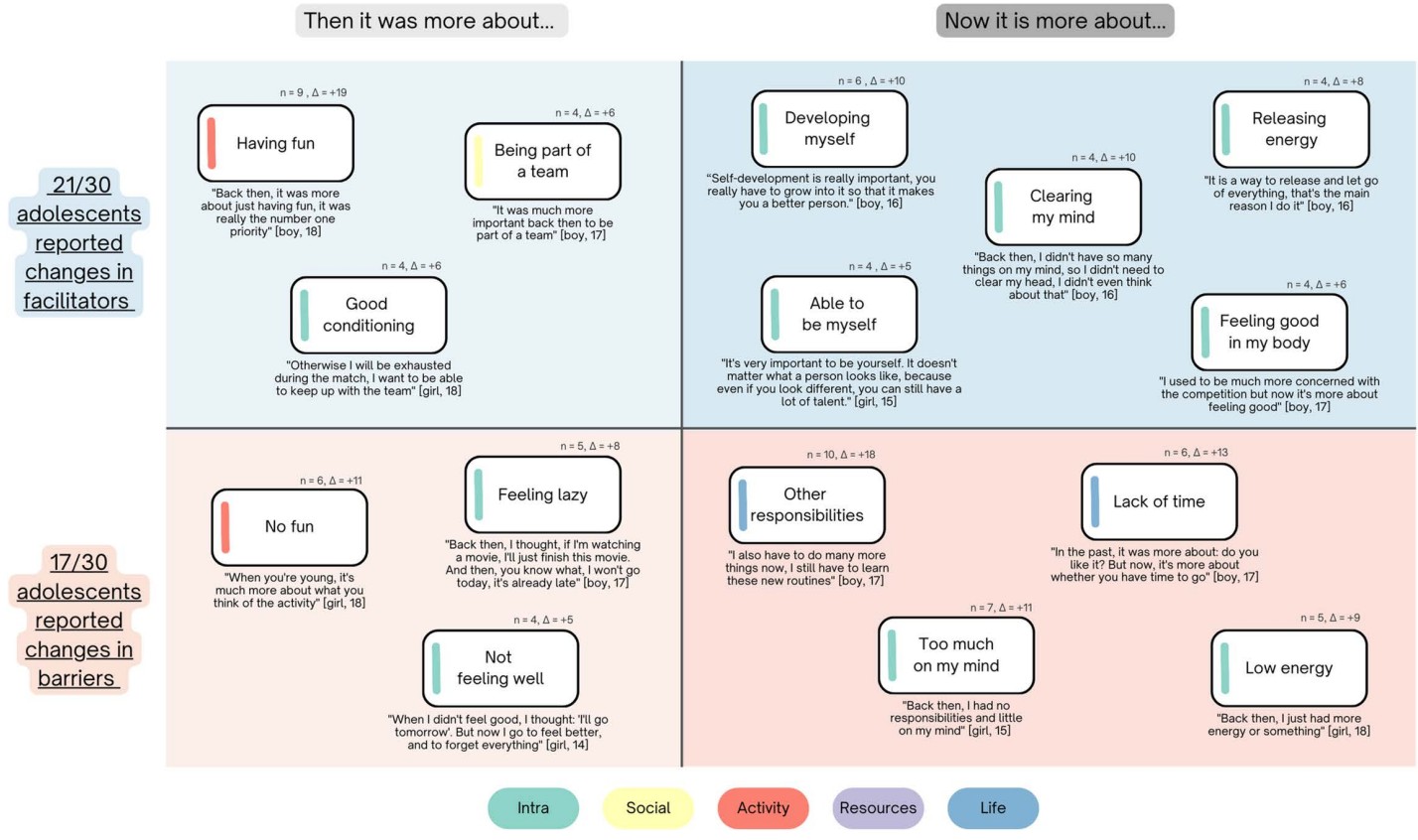

**Fig 4. The largest perceived shifts in barriers and facilitators for physical activity maintenance between past and present experiences.** Δ indicates change in ranked score.

In the past, enjoyment stood out as a primary motivator for engaging in PA for the participants. Additionally, having a good physical conditioning and being a part of a team and were important facilitators in the past. In contrast, present facilitators were more self-directed, including both the ability to be true to themselves and to pursue their self-development. Moreover, PA has turned into a mechanism that can regulate both mental and physical wellbeing such as mental clarity and energy levels.

Past barriers centered more on the activity itself or adolescents' physical and mental states, with the prospect of not enjoying the activity as the primary barrier for PA engagement. In addition, feelings of personal discomfort and moments of laziness were common obstacles. In contrast, present barriers predominantly revolved around external factors and obligations. Adolescents now grapple with balancing responsibilities, which contributes to feelings of mental overload and a perceived depletion of time and energy from doing other tasks.

## Discussion

This study employed a mixed methods approach combining interviews and card sorting based on Q-methodology to study how adolescent experience their PA maintenance. For our first research aim, we identified barrier and facilitators profiles which represent different adolescent perspectives on PA maintenance. For our second research aim, we identified how barriers and facilitators for PA were perceived to change over time.

## Barrier & facilitator profiles

Factor analysis on the card sorting task resulted in five facilitator and four barrier factors, showing that adolescents' experienced barriers and facilitators to PA maintenance are not uniform but can be grouped into distinct profiles. Profile A (*Mental and physical health benefits*) and B (*A way to be myself around others*) were grounded in the direct, immediate characteristics of the activity. Profile A emphasized fun and direct physical and mental health outcomes, while B, the most distinct factor and composed exclusively of girls, highlighted the importance of social context and support. In contrast, profiles C, D, and E were more goal-driven and future-oriented. Profile C prioritized health goals and performance milestones, D emphasized developing strength and confidence, and E combined forward-looking self-development with enjoyment. However, our factor solution did not capture all variation, as ten participants did not fit any facilitator factor, indicating additional unrepresented perspectives exist beyond these profiles.

The barrier profiles showed less overlap than the facilitator profiles, making them more distinct. They differed in the degree to which they emphasized internal versus external influences. Profile A was positioned firmly at the internal end of the spectrum, focusing almost exclusively on barriers such as low motivation, limited energy, or feelings of laziness. At the opposite end, profile D centered almost entirely on environmental obstacles unrelated to the activity's content (e.g., weather, travel distance). Profiles B and C combined internal and external barriers. Profile C connected external life demands to depleted internal resources, while Profile B emphasized the fit between the activity and the individual (e.g., questioning whether the activity feels right or safe).

Together, these findings indicate that adolescents' PA maintenance experiences are shaped by multiple influences and can be grouped into distinct profiles. This is consistent with prior work showing that youth construct their own meanings, values, and perceptions around health behaviors, and that these subjective configurations shape maintenance [19,20]. However, the profiles we identified are unlikely to represent the full spectrum of adolescent experiences. The explained variance of the factor solutions was relatively low (below 60%), and could be explained by other contributors to maintenance, such as self-regulation skills, habit strength, and environmental resources [13], were only indirectly captured in our card set. Although most profiles emphasize intrapersonal influences, adolescents' accounts also point to external demands from the environment, showing the interaction between context individual characteristics.

## Perceived changes over time

Our results show that adolescents perceived both barriers and facilitators for PA to change over time. In the past, facilitators were more tied to the direct characteristics of the activity (e.g., team atmosphere, having fun, and feeling physically capable). Over time, these changed towards more "higher order" goals related to self-development and self-expression, for example, gaining benefits beyond the context of the activity and becoming who they want to be. This change suggests that earlier PA engagement depends more on immediate, experiential qualities, whereas current engagement (among our sample of PA maintainers for at least two years) is supported by motives more aligned with identity and self-directed goal pursuit. Viewed through a self-determination theory lens, earlier motivation appeared to be connect more to relatedness and competence needs (e.g., connection in a team setting and feeling capable), whereas current facilitators increasingly reflect autonomy (developing myself, able to be myself). Furthermore, adolescents also seem to develop a broader conception of wellbeing (energy balance, feeling good, clearing my mind), suggesting that their understanding of what PA affords now also includes psychological and self-regulatory benefits, not just physical ones.

Barriers followed a complementary pattern. Earlier in adolescence, obstacles tended to be more internal and immediate (e.g., boredom, low mood, or not feeling well). In contrast, present barriers were more often due to new external demands, driven by increasing responsibilities from school, work and family, and consequently by reduced available time or energy for PA. This shift reflects a broader life context in which adolescents constrained not only by internal factors such as low motivation or boredom, but also by the need to juggle multiple demands from their surroundings.

The changes in barriers and facilitators echo findings from other studies and align with developmental psychology theories [17,19,22]. Adolescence is a period of identity exploration and formation, occurring alongside new academic, familial, and social responsibilities and interests [30,31]. These transitions require continual reprioritization and trade-offs in how limited time and cognitive resources are allocated. Our sample of maintainers appears to counteract these life demands by drawing increasingly on motivation grounded in autonomy and positive outcomes, which can contribute to their sustained engagement. Notably, while most participants reported changes, some reported no change in their barriers (n = 12) or facilitators (n = 9), indicating that these shifts are not universal. Participants who reported changes over time were one year older on average, suggesting that such shifts may become more likely as adolescents progress along their developmental pathways. However, the age ranges of changers and non-changers overlapped substantially (ages 13–16 vs. 13–18), indicating that age alone does not fully explain these differences. Moreover, across the sample, "having fun" remained the most consistent facilitator, underscoring that immediate enjoyment continues to be a core motivator for PA, even as other motives evolve.

The primary strength of this research is the investigation of subjective perspectives and experiences of adolescents around maintained PA. Our findings have several implications for promoting sustained physical activity from a person-based developmental perspective. First, interventions should treat adolescents as evolving agents, not static targets. Priorities, motivations, and constraints change over time, and simply encouraging more physical activity among adolescents without acknowledging competing demands can increase friction rather than support engagement. Second, because enjoyment remains a core driver, especially earlier in PA engagement, PA should be framed not merely as a health imperative but primarily as an intrinsically enjoyable experience, a point often secondary in health promotion programs. Third, PA interventions must be adaptive and reflective. Although the immediate experiences of the physical activity are most important, for maintained PA, other goals also become important such as self-development, regulating energy balance, and feeling like you can be yourself. These more future-oriented influences should be given space to develop within PA interventions and actively encouraged, for example by embedding periodic reflection opportunities to reassess how PA also contributes to other goals. As several participants commented: *'I knew all these things already, but it is good to get an overview of it like this.'* Health promoting programs could provide adolescents with tools to create an overview of their barriers and facilitators can help guide autonomous decision-making as their needs evolve with shifting life contexts, supporting them in renegotiating how PA fits alongside growing responsibilities and changing identities.

## Limitations & future research

This study is not without limitations. First, the study took place in a Dutch adolescent population in an urban area of the Netherlands, which comes with culture-specific considerations at both contextual and developmental levels. The Netherlands is a high-income country with relatively good access to PA opportunities, but most adolescents don't meet the recommended amount of PA [32]. Furthermore, none of the researchers were adolescents, so the selection of the stimuli used during interviews, the creation of the card set, and the interpretation of interviews necessarily reflect adult perspectives. Our assumptions and consequent study design decisions may have shaped participants' responses and may not fully capture how adolescents themselves conceptualize PA. These points make the generalizability of these research harder to other contexts. Future research could involve adolescents as co-researchers and by conducting similar studies in different cultural contexts. Second, we are unable to link our findings to actual PA data of our participants (e.g., accelerometer measures or activity logs). While engaging in one physical activity for at least two years was an inclusion criteria of the study, but we did not measure actual PA levels during the data collection. Consequently, we cannot determine how perceived changes in barriers and facilitators relate to actual PA engagement or predict long-term adherence. Future research could integrate objective PA data with experiential insights to examine how subjective experiences interact with measured activity levels. Third, we compared changes across a wide age range (13–18 years). Within this range, developmental stages differ substantially in cognitive, social, and physical ways. Future studies could focus on narrower age ranges to clarify how developmental transitions influence PA maintenance profiles.

## Conclusions

This study provides valuable insights into adolescents' PA maintenance experiences by identifying distinct profiles of barriers and facilitators and showing how these can change over time. It highlights that adolescents' experiences are diverse and dynamic, with differences linked to internal versus external influences and to immediate versus more distal characteristics of PA. By capturing these patterns, our research deepens understanding of lived experiences around PA maintenance in adolescence. These findings point to new directions for designing tailored, person-based interventions that account for developmental changes and support sustainable PA among adolescents.

## Supporting information

**S1 Fig. Materials used during interviews to sensitize adolescents about their active lifestyle.**
(PDF)

**S2 Table. Final list of facilitators and facilitators used during Q-sorting.**
(PDF)

**S3 Table. The different types of physical activities that adolescents had maintained.**
(PDF)

**S4 Table. Weighted average z-scores per facilitator and barrier factor.**
(PDF)

## Acknowledgments

We would like to thank S. Loos and S. Hondmann for their time and energy in helping with the data collection process.

## Author contributions

**Conceptualization:** Timothy A. Houtman, Froukje Sleeswijk Visser, Valentijn Visch.

**Data curation:** Timothy A. Houtman.

**Formal analysis:** Timothy A. Houtman.

**Funding acquisition:** Valentijn Visch.

**Investigation:** Timothy A. Houtman.

**Methodology:** Timothy A. Houtman.

**Project administration:** Timothy A. Houtman.

**Supervision:** Froukje Sleeswijk Visser, Amy van Grieken, Valentijn Visch.

**Validation:** Timothy A. Houtman.

**Visualization:** Timothy A. Houtman.

**Writing – original draft:** Timothy A. Houtman.

**Writing – review & editing:** Timothy A. Houtman, Froukje Sleeswijk Visser, Amy van Grieken, Valentijn Visch.

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
