## [Decision Letter · Decision Letter 0]

19 May 2025

PONE-D-24-54299Dynamics and Diversity of Adolescent Perceptions of Barriers and Facilitators for Physical Activity MaintenancePLOS ONE

Dear Dr. Houtman,

Thank you for submitting your manuscript to PLOS ONE. After careful consideration, we feel that it has merit but does not fully meet PLOS ONE’s publication criteria as it currently stands. Therefore, we invite you to submit a revised version of the manuscript that addresses the points raised during the review process. 

We look forward to receiving your revised manuscript.

Kind regards,

Hidetaka Hamasaki

Academic Editor

PLOS ONE

Journal Requirements:

“This research was funded by a grant awarded to dr. V.T. Visch from the Dutch Research Council (NWO). File Number: 403.19.246”

4. Please note that funding information should not appear in the Acknowledgments section or other areas of your manuscript. We will only publish funding information present in the Funding Statement section of the online submission form. Please remove any funding-related text from the manuscript. 

5. We note that you have indicated that there are restrictions to data sharing for this study. For studies involving human research participant data or other sensitive data, we encourage authors to share de-identified or anonymized data. However, when data cannot be publicly shared for ethical reasons, we allow authors to make their data sets available upon request. For information on unacceptable data access restrictions, please see http://journals.plos.org/plosone/s/data-availability#loc-unacceptable-data-access-restrictions. 

Reviewers' comments:

Reviewer's Responses to Questions

**Comments to the Author**

1. Is the manuscript technically sound, and do the data support the conclusions?

Reviewer #1: Yes

Reviewer #2: Yes

Reviewer #3: Yes

Reviewer #4: Yes

2. Has the statistical analysis been performed appropriately and rigorously? 

Reviewer #1: Yes

Reviewer #2: Yes

Reviewer #3: Yes

Reviewer #4: Yes

3. Have the authors made all data underlying the findings in their manuscript fully available?

Reviewer #1: No

Reviewer #2: Yes

Reviewer #3: No

Reviewer #4: No

4. Is the manuscript presented in an intelligible fashion and written in standard English?

Reviewer #1: Yes

Reviewer #2: Yes

Reviewer #3: Yes

Reviewer #4: Yes

5. Review Comments to the Author

Reviewer #1: This is an interesting piece of work, using interesting methods of engaging with young people. The exploration of PA maintenance is important. Below are some more specific comments I think need to be addressed to strengthen the manuscript. I have selected major revision purely because some points may need some time and consideration to be addressed. Especially regarding clarity and including important information and considerations I think are currently missing. I have uploaded a document with specific points to be addressed.

Reviewer #2: I would like to appretiate the efforts of the authors in implementing the project and writing this article “Dynamics and Diversity of Adolescent Perceptions of Barriers and Facilitators for Physical Activity Maintenance”.

The aim of this very study is The aim is to understand how barriers and facilitators for PA maintenance change during adolescence. The second aim is to examine how these barriers and facilitators act within adolescents and how they differ between them.

The study is very interesting and beneficial for practice.

I have the following comments and questions:

Methods:

- The methodology is innovative.

- I recommend adding age ± SD years

- It would be useful to provide data on the socioeconomic status or ethnicity of the participants.

- It is not clear from the methodology how the authors arrived at the claim of "development with age". This claim seems speculative without longitudinal data.

- For better transparency it would be useful to specify the cluster analysis procedure (algorithm, validation metrics).

Discussion:

- I recommend discussing the effect of the Covid-19 pandemic on the results

Reviewer #3: I read this paper with great interest. This is an ambitious study that deeply explores the dynamics and diversity of perceptions regarding barriers and facilitators to physical activity (PA) maintenance during adolescence using a mixed-methods approach. The approach of addressing the crucial issue of adolescent PA maintenance from two perspectives—'temporal changes' in factors and 'inter-individual diversity'—is highly commendable. This perspective is likely to contribute to the development of more effective interventions. Furthermore, the mixed-methods approach combining exploratory interviews and Q-methodology-based card sorting is appropriate for capturing the complex perceptions and experiences of individuals from multiple angles. Moreover, the discussion links these findings to adolescent developmental characteristics and prior research, making the content rich in both theoretical and practical implications. Below, I offer some suggestions for further improvement.

Results:

Regarding the description 'most notable shifts' in the 'Changing dynamics over time' subsection of the Results section, clarifying the specific criteria for 'notability' (e.g., items reported as changed by a larger number of participants?) within the text would enhance reader understanding.

Methods:

In the 'Q-set development' subsection of the Methods section, the developed barriers (31 items) and facilitators (36 items) are mentioned. To help readers grasp the study's content without needing to refer to the appendix, perhaps consider providing a few representative examples of these items within the main text.

Discussion:

The limitation regarding the lack of consideration for the influence of the 'type of activity' being maintained is mentioned, and I believe this is a very important point. It might be beneficial to expand the discussion or speculation on how the identified clusters and temporal changes could potentially be influenced by differences in activity types, such as individual vs. team sports, or structured sports vs. free-form exercise. This would also be valuable for making the implications for future research more concrete.

While Q-methodology allows for capturing the diversity of perspectives even with relatively small samples, from the perspective of the stability and generalizability of the cluster analysis results, it might be appropriate to mention the sample size as a limitation.

Reviewer #4: PONE-D-24-54299: Dynamics and Diversity of Adolescent Perceptions of Barriers and Facilitators for Physical Activity Maintenance

The title clearly reflects the content and purpose of the article. The abstract clearly summarizes the background, purpose, method, and main findings of the study. Practical implications of the results are also included in the abstract; this allows the reader to see the application area of the research.

In order for the summary to be more effective, the names and basic characteristics of the “five different facilitator clusters and four barrier clusters” presented can be briefly included, albeit superficially. In addition, the number of participants and age range can be presented more concisely in a single sentence in the method section.

Introduction

Current literature and research needs on the subject are clearly stated. The difficulties of maintaining physical activity behavior and current deficiencies are emphasized. The problem definition and the objectives of the article are well structured. The study effectively identified an important gap in the existing literature on physical activity maintenance in adolescents. The argument that physical activity maintenance behavior is different from initiation behavior is strongly supported. The literature review is comprehensive and integrates relevant theoretical frameworks and previous studies well. An understanding of the developmental stages of adolescents is successfully integrated into the conceptual framework of the study.

In the introduction, the gaps in the existing literature can be supported with more concrete data (statistical values or examples). In addition, the distinction between “behavioral maintenance” and “behavioral initiation” could be given in a clearer.

The research objectives could be stated in a clearer and more structured manner. In particular, more specific information could be provided on how to examine change over time and individual differences.

An operational definition for the concept of physical activity “maintenance” should be provided. The authors asked participants to recall “a physical activity that they have done for at least 2 years and are still doing,” but it is not explained how this definition of maintenance relates to the literature.

The importance of the person-centered (idiographic) approach is emphasized, but more theoretical background on this concept could be provided.

Method

Research Design

The selection of a mixed method approach is appropriate for the complex nature of the subject. Both qualitative (interviews) and quantitative (Q methodology and card sorting) applications are explained in detail.

Participant selection criteria and the sampling process can be specified in more detail. In particular, providing information about the socioeconomic characteristics of the school(s) where the interviews were conducted or the diversity in the sample is important for the generalizability of the study.

Data Collection Tools and Procedure

The creative techniques used in the interviews (such as object/character selection and sorting) are innovative and increase participant interaction. The study adopted a mixed methodological approach that successfully integrated both qualitative and quantitative methods. The two-stage research design (exploratory interviews and card sorting) is well structured.

The use of Q methodology is an appropriate choice for understanding the subjective perspectives of individuals.

The data collection process and ethical procedures are explained in detail.

Examples of the cards used and the selection process (reference to the “appendix” files should be provided, but a brief explanation should be added in the main text) can be presented as a summary in the main text. Replication would be easier if the reason and how the cards were selected were explained more clearly.

More information about the sample selection and sample size determination process would be useful.

More methodological details could be provided about the cluster analysis used in the data analysis process.

More information about the demographic characteristics of the participants (socioeconomic status, ethnicity, etc.) would be useful to assess the generalizability of the results.

The analytical steps in the transition from exploratory interviews to the card sorting activity could be more clearly stated.

Results

The most frequently highlighted barriers and facilitators are presented clearly and in a tabular form. How change occurs with age or over time is presented in a clear manner.

The findings are presented in a well-organized manner that is consistent with the research objectives.

The analysis of changes over time provides important insights into understanding developmental changes in adolescents’ motivation to pursue physical activity.

The cluster analysis successfully reveals different perspectives on adolescents’ motivation to pursue physical activity.

The detailed descriptions of the clusters for both facilitators and barriers contribute to the depth of presentation and analysis.

A more comprehensive visualization of the five facilitators and four barrier clusters obtained from the cluster analysis may help interpret the results.

Additional analyses can be provided on the relationship of the clusters to demographic characteristics, types of physical activity, or other relevant variables.

Figure/table references (e.g. Figure 2) should be explained further in the text. The reader should be provided with brief explanations of what the key tables and figures support in the text.

The relationship or interaction between barriers and facilitators can be examined in more detail.

There is no clear time frame defined for "past" and "present", which can make it difficult to interpret changes over time.

Clustering

The clustering for both barriers and facilitators provides a good picture of the diversity of young people's experiences. The basic characteristics/criteria and representative views of each cluster are explained in detail.

The number of people in the clusters and the distribution of the clusters should be given more clearly in numerical terms. In some clusters, gender/etc. characteristics (e.g. girls' group, boys' group) should be emphasized but are omitted.

Discussion

The results obtained are interpreted by comparing them with the existing views in the literature. The practical implications of the results and personalized intervention suggestions are valuable. The study limitations are honestly addressed and directions for future research are offered. Practical implications for developing interventions are presented. The study successfully highlights the dynamic and diverse nature of physical activity maintenance behavior in adolescents.

The limitations regarding the generalizability of the sample covered by the study (local/Dutch context) should be discussed more clearly. In addition, the limitations of the study include data types/distribution, risk of bias, small sample size and weaknesses of the Q methodology, which can be discussed with examples.

The discussion could be strengthened by a more comprehensive situating of physical activity maintenance behavior in light of different theoretical frameworks.

More emphasis could be placed on the implications of the study for the broader public health and behavior change arena.

More specific and actionable recommendations could be provided for intervention design.

Further discussion could be provided on the generalizability of the findings of this study among adolescents in the Netherlands to different cultural contexts.

Conclusion and Contributions

The conclusion is concise; implications of the findings for interventions for youth are clearly stated.

Recommendations for policy makers and practitioners can be summarized in short bullet points.

References

References are up-to-date and consistent with the literature in the field. Ethics statement, funding, and conflict of interest statements are properly included.

References contain some minor spelling/formatting errors; careful review for adherence to standard journal style.

Writing and Organization

The article is well structured and clearly written.

References are up-to-date and relevant.

Tables and figures are well integrated with the findings.

The abstract effectively summarizes the main components of the study.

Aspects that Need Improvement The language flow could be improved and more concise expressions could be used.

The data collection and analysis processes could be more clearly separated by using more subheadings in the Methodology section.

The legends of the figures could be more detailed.

Additional visual elements could be added to the Results section to better illustrate the results from the cluster analysis.

General Assessment and Recommendations

This article is an innovative study that provides valuable insights into adolescents’ physical activity maintenance behaviors. The mixed methodology approach and the use of Q methodology contribute significantly to the in-depth understanding of the topic. The findings on how adolescents’ motivation to maintain physical activity changes over time are particularly striking and have important implications for intervention design.

Considering the points that need to be corrected above, the overall quality of the article will further improve and provide a stronger foundation for the development of effective interventions for promoting physical activity in adolescents. In particular, strengthening the methodological explanations, presenting the findings with richer visualizations, and establishing stronger connections to theoretical frameworks will increase the impact of the article.

6. PLOS authors have the option to publish the peer review history of their article (what does this mean? ). If published, this will include your full peer review and any attached files.

**Do you want your identity to be public for this peer review?** For information about this choice, including consent withdrawal, please see our Privacy Policy .

Reviewer #1: No

Reviewer #2: No

Reviewer #3: No

Reviewer #4: **Yes: ** Cihan Aygün

---

## [Author Response · Author response to Decision Letter 1]

14 Aug 2025

Dear reviewer and editor,

Thanks for you effort and time. We think the manuscript has significantly improved by incorporating the feedback. The made changes are described and detailed and the newly uploaded documents.

---

## [Decision Letter · Decision Letter 1]

10 Sep 2025

Dynamics and Diversity in Adolescents’ Experienced Barriers and Facilitators for Physical Activity Maintenance

PONE-D-24-54299R1

Dear Dr. Houtman,

We’re pleased to inform you that your manuscript has been judged scientifically suitable for publication and will be formally accepted for publication once it meets all outstanding technical requirements.

Kind regards,

Hidetaka Hamasaki

Academic Editor

PLOS ONE

Additional Editor Comments (optional):

Reviewer #3:

Reviewer #4:

Reviewers' comments:

Reviewer's Responses to Questions

**Comments to the Author**

1. If the authors have adequately addressed your comments raised in a previous round of review and you feel that this manuscript is now acceptable for publication, you may indicate that here to bypass the “Comments to the Author” section, enter your conflict of interest statement in the “Confidential to Editor” section, and submit your "Accept" recommendation.

Reviewer #3: All comments have been addressed

Reviewer #4: All comments have been addressed

2. Is the manuscript technically sound, and do the data support the conclusions?

Reviewer #3: Yes

Reviewer #4: Yes

3. Has the statistical analysis been performed appropriately and rigorously? 

Reviewer #3: Yes

Reviewer #4: Yes

4. Have the authors made all data underlying the findings in their manuscript fully available?

Reviewer #3: Yes

Reviewer #4: No

5. Is the manuscript presented in an intelligible fashion and written in standard English?

Reviewer #3: Yes

Reviewer #4: Yes

6. Review Comments to the Author

Reviewer #3: I have reviewed the revised manuscript.

I find that the quality of the paper has been significantly improved through the authors' revisions. I recommend that the manuscript be accepted for publication.

As a note for the future, the response letter appeared to group comments from multiple reviewers, which made it somewhat difficult to follow the responses to individual points. For future submissions, responding to each reviewer's comments individually in the response letter would contribute to a smoother and more constructive review process.

Reviewer #4: The authors carefully addressed all comments and criticisms from the peer reviewer and made comprehensive improvements in the methodology, analysis, and discussion sections. The results are clearly presented, the discussion is well-aligned with the literature, and the limitations are thoroughly described.

7. PLOS authors have the option to publish the peer review history of their article (what does this mean? ). If published, this will include your full peer review and any attached files.

**Do you want your identity to be public for this peer review?** For information about this choice, including consent withdrawal, please see our Privacy Policy .

Reviewer #3: No

Reviewer #4: **Yes: ** Cihan Aygün

---

## [Editor Report · Acceptance letter]

PONE-D-24-54299R1

PLOS ONE

Dear Dr. Houtman,

I'm pleased to inform you that your manuscript has been deemed suitable for publication in PLOS ONE. Congratulations! Your manuscript is now being handed over to our production team.

Kind regards,

on behalf of

Dr. Hidetaka Hamasaki

Academic Editor

PLOS ONE